# A Celastrol Drug Delivery System Based on PEG Derivatives: The Structural Effects of Nanocarriers

**DOI:** 10.3390/molecules28031040

**Published:** 2023-01-20

**Authors:** Yansong Zhang, Lijuan Ding, Ting Wang, Xiangtao Wang, Bo Yu, Fei Jia, Meihua Han, Yifei Guo

**Affiliations:** 1College of Food and Drug, Luoyang Normal University, Luoyang 471934, China; 2Institute of Medicinal Plant Development, Chinese Academy of Medical Sciences & Peking Union Medical College, No. 151, Malianwa North Road, Beijing 100193, China; 3Key Laboratory of New Drug Discovery Based on Classic Chinese Medicine Prescription, Chinese Academy of Medical Sciences, No. 151, Malianwa North Road, Beijing 100193, China; 4Key Laboratory of Bioactive Substances and Resources Utilization of Chinese Herbal Medicine, Ministry of Education, Chinese Academy of Medical Sciences & Peking Union Medical College, Beijing 100193, China

**Keywords:** structural effect, degree of branching, composition, hydrophobic interaction, steric hindrance

## Abstract

The therapeutic efficacy of nanoscale drug delivery systems is related to particle size, zeta potential, morphology, and other physicochemical properties. The structure and composition of nanocarriers may affect their physicochemical properties. To systematically evaluate these characteristics, three analogues, namely polyethylene glycol (PEG), PEG-conjugated octadecylamine (PEG-C18), and tri(ethylene glycol) (TEG), were explored as nanocarriers to entrap celastrol (CSL) via the injection-combined dialysis method. CSL nanoparticles were successfully prepared as orange milky solutions, which revealed a similar particle size of approximately 120 nm, with narrow distribution and a negative zeta potential of −20 mV. All these CSL nanoparticles exhibited good storage stability and media stability but presented different drug-loading capacities (DLCs), release profiles, cytotoxicity, and hemolytic activity. For DLCs, PEG-C18/CSL exhibited better CSL entrapment capacity. Regarding the release profiles, TEG/CSL showed the lowest release rate, PEG-C18/CSL presented a moderate release rate, and PEG/CSL exhibited a relatively fast release rate. Based on the different release rates, PEG-C18/CSL and TEG/CSL showed higher degrees of cytotoxicity than PEG/CSL. Furthermore, TEG/CSL showed the lowest membrane toxicity, and its hemolytic rate was below 20%. These results suggest that the structural effects of nanocarriers can affect the interactions between nanocarriers and drugs, resulting in different release profiles and antitumor activity.

## 1. Introduction

Nanoscale drug delivery systems (NDDS) play important roles in hydrophobic drug delivery [1,2]; therefore, they have been extensively studied during the last several years [3,4]. Many nanomaterials, especially amphiphilic block copolymers, are utilized as nanocarriers to load hydrophobic drugs via physical entrapment [5,6,7]. In this process, the hydrophobic chains in amphiphilic copolymers entrap hydrophobic drugs, which form the inner core, while the hydrophilic chains form the outer shell, thus contributing to aqueous solubility [8,9]. During the formation of these amphiphilic copolymers, PEGs are applied as hydrophilic portions owing to their excellent aqueous solubility, biosafety, and stealthy properties [10,11].

Although amphiphilic copolymers have shown efficacy in the formation of NDDS, their drug-loading content, stability, release profile, and antitumor activity are significantly affected by the structure of nanocarriers [12,13]. It is reported that the structure or components of amphiphilic copolymers can influence their particle size, surface charge, morphology, and other physicochemical properties, thus inducing different therapeutic effects [14,15,16]. In a previous study, PEG was conjugated with different numbers of PCLs to form amphiphilic copolymers, and a fiber-like and spherical morphology was observed [17]. Moreover, nanocarriers with a high degree of branching exhibit better antitumor efficacy owing to their unique topological structure and more functional groups [18,19,20]. In another study, to research the effect of composition, different PEGylated amphiphilic copolymers were explored to construct NDDS [21]; however, it is difficult to reach convincing conclusions on the influence of amphiphilic copolymers on antitumor efficacy due to the lack of a unified standard.

In our previous research, the structural effect and components of nanocarriers were studied in detail. Different from the use of amphiphilic copolymers as nanocarriers, the potential of hydrophilic polymers to entrap hydrophobic drugs via several PEG derivatives was investigated, and it was found that the structures of nanocarriers affect the particle size and morphology of NDDS, further resulting in different levels of antitumor activity [22]. The results show that the tumor inhibition rate depends on the degree of the branching of nanocarriers, which is significantly enhanced with a change in the structure from a linear chain to a branched dendron [23]. Furthermore, it is clear that the components of nanocarriers also affect their antitumor activity. Although hydrophilic PEGs as nanocarriers have shown good entrapment capacity to load hydrophobic drugs and prepare effective NDDS, they have lower levels of antitumor efficacy than PEG-decorated amphiphilic copolymers, which have stronger interactions with hydrophobic drugs or cell membrane [24,25]. These results reveal that the antitumor activity of NDDS based on PEGs as nanocarriers may be promoted by increasing the degree of the branching of PEG chains or introducing hydrophobic chains. While these effects have been individually researched in previous reports, it is difficult to reach conclusions on the combination of these effects based on the published results.

To systematically compare and study the effects of the structure and components of nanocarriers, in this study, three different PEG derivatives were applied as nanocarriers to prepare NDDS, in which hydrophobic celastrol was selected as a model drug. The injection-combined dialysis method was utilized to prepare CSL-loaded nanoparticles. After the successful preparation of nanoparticles, their particle size, size distribution, zeta potential, stability, release profile, antitumor efficacy in vitro, and hemolytic capacity were evaluated.

## 2. Results and Discussion

### 2.1. Celastrol-Loaded Nanoparticles

To compare the effects of the structure and hydrophobic chains, PEG, PEG-C18, and TEG were selected as nanocarriers to construct CSL-loaded nanoparticles via the ultrasonication–dialysis method (Figure 1). CSL and the nanocarriers (8/1, *w*/*w*) were dissolved in DMF, injected into distilled water under continuous ultrasonication, and dialyzed against deionized water. Finally, orange CSL nanoparticle solutions were successfully obtained.

The DLCs of these CSL nanoparticles were determined via HPLC. All the nanocarriers presented moderate to good encapsulated efficacy, and the DLCs were 77.8%, 87.3%, and 68.5% for PEG/CSL, PEG-C18/CSL, and TEG/CSL, respectively (Table 1). The relevant encapsulate efficiencies (EE) where it was considered that CSL NPs presented “core-shell” structure, in which CSL was encapsulated by nanocarriers to form the hydrophobic core, meanwhile, the hydrophilic portion of nanocarriers dispersed on the outer of nanoparticles to contribute to the hydrophilicity. During the assembly process, a DLC would be affected by the hydrophobic interactions and steric hindrance. PEG and PEG-C18 revealed a similar degree of entrapped efficacy, which was better than that of TEG. This phenomenon could be attributed to the different degrees of steric hindrance; PEG and PEG-C18 have flexible linear chains and, therefore, low steric hindrance. Although the structure of TEG is also a linear chain, many tri(ethylene glycol) compounds exist as side chains around the main chain, thus enhancing steric hindrance and reducing the space capacity of the nanocarriers, which induced low drug-loading capacity [26]. Moreover, PEG-C18 presented higher entrapped efficacy than PEG, because PEG-C18 presented more hydrophobicity than PEG. C18 aliphatic chains in PEG-C18 enhance the hydrophobicity of nanocarriers, which results in stronger hydrophobic interactions between PEG-C18 and CSL. The enhanced drug-loading capacity of PEG-C18 is shown in Table 1.

### 2.2. Particle Size and Morphology of CSL Nanoparticles

All these CSL-loaded nanoparticles were measured via DLS, and their hydrodynamic diameter, particle size distribution, and zeta potential are shown in Table 1. The three CSL nanoparticles presented similar particle sizes, PDIs, and zeta potentials. The hydrodynamic diameters of these CSL nanoparticles were approximately 120 nm, and a single peak was observed in DLS curves with a narrow distribution (Figure 2a,c,e). The PDI values of these nanoparticles were approximately 0.1. These results suggest that the CSL nanoparticles exhibited good uniformity. Moreover, the zeta potentials of these CSL nanoparticles were approximately −23 mV due to the carboxyl group of CSL. A high absolute potential value ensures the better stability of nanoparticles in an aqueous solution.

The morphologies of these CSL nanoparticles were observed using TEM, which revealed similar nanospheres (Figure 2b,d,f). For all the nanoparticles, the particle sizes measured using TEM were smaller than those measured via DLS, which were approximately 80 nm. This phenomenon could be explained by the different measuring conditions of these two techniques. The particle size detected with DLS is the hydrodynamic diameter, while in TEM, it is the measured size of the dry nanoparticles.

### 2.3. Stability

These CSL nanoparticles were stored at 4 and 25 °C for 28 days to determine their storage stability, and their particle sizes were recorded and are shown in Figure 3. No aggregation or precipitation was observed during the entire storage period at 25 °C. Their hydrodynamic diameters were maintained at approximately 125 nm. When the nanoparticles were stored at 4 °C, a similar phenomenon occurred. These results reveal that CSL nanoparticles have excellent storage stability.

The media stabilities were also assessed. The CSL nanoparticles were added in PBS (pH 7.4), 5% glucose solution, 0.9% solution, and plasma. All the CSL nanoparticles presented good levels of stability in PBS (pH 7.4), 5% glucose solution, and plasma. After mixing with the 5% glucose solution, the particle sizes slightly increased, which were approximately 155, 175, and 185 nm for PEG/CSL, TEG/CSL, and PEG-C18/CSL, respectively. During the subsequent 24 h, the particle sizes were maintained, and no significant change was observed during the incubation period (Figure 4a). A similar phenomenon occurred when the CSL nanoparticles were incubated with plasma: The particle sizes of these nanoparticles slightly fluctuated within the initial 4 h and then maintained a constant in the following 20 h (Figure 4b). This phenomenon may be attributed to the interactions between nanoparticles and plasma proteins [27,28]. It was reported that nanoparticles might be regulated by plasma proteins after entering a physiological environment, which could bind the surfaces of the nanoparticles to form the protein crown. The CSL nanoparticles interacted with plasma proteins to form nanoparticle–protein complexes within the initial 4 h, resulting in the fluctuation in the particle size; the complexes were stable in the subsequent incubation time, and the particle sizes were maintained.

### 2.4. Cumulative Release Behavior

To determine the release profiles, the dialysis membrane method was used with PBS (pH 7.4) as the release medium to mimic the release process in a physiological environment. The free CSL under the same conditions was used as the control (Figure 5). The free CSL (the CSL DMSO solution) presented a fast release rate, as it was completely released in 12 h. CSL nanoparticles exhibited sustaining release profiles, and their release process ranged from 4 to 8 days. Compared with the free CSL, the CSL nanoparticles revealed slow release rates, which may be attributed to the core–shell structure of the CSL nanoparticles, consistent with previous reports [29,30].

Moreover, these CSL nanoparticles presented similar release tendencies but different release rates. Specifically, more than 95% of the CSL was released from PEG/CSL, PEG-C18/CSL, and TEG/CSL nanoparticles at 4, 6, and 8 days, respectively. PEG/CSL exhibited a fast release rate, PEG-C18/CSL had a moderate release rate, and TEG/CSL presented a relatively slow release rate. This result was caused by the different interactions and extent of steric hindrance among these CSL nanoparticles. PEG/CSL nanoparticles presented faster release rates than the other two nanoparticles because the hydrophobic interactions between PEG and CSL were weak, resulting in their fast release. For PEG-C18/CSL nanoparticles, the hydrophobic interactions were enhanced due to the hydrophobicity of C18 chains. For TEG/CSL nanoparticles, although the hydrophobicity was almost the same as that in PEG, the steric hindrance was stronger than the steric hindrance in PEG. After entrapment by TEG and the formation of the hydrophobic core, the SL release rate from the inner core to the outer shell was significantly affected by the steric hindrance of the outer shell. Based on these results, the hydrophobicity and steric hindrance of nanocarriers may affect the drug release rate [31,32]. Therefore, these factors should be thoroughly considered when a drug delivery system is prepared.

### 2.5. MTT Assay

A 4T1 cell line was utilized to evaluate the cytotoxicity of CSL nanoparticles, and a CSL DMSO solution under the same conditions was used as the control (Figure 6). All these samples showed concentration-dependent cytotoxicity, and the IC_50_ values were 1.55 ± 0.21, 0.96 ± 0.08, 0.72 ± 0.04, and 0.70 ± 0.06 μg/mL for the free CSL, PEG/CSL, PEG-C18/CSL, and TEG/CSL nanoparticles, respectively. Compared with the free CSL, the cytotoxicity of the CSL nanoparticles was significantly enhanced (*p* < 0.001). This finding is consistent with the findings of previous reports [33], which could be explained by the different uptake mechanisms of free drugs and nanoparticles. Free drugs cross the cell membrane via passive diffusion, while nanoparticles transfer to cells via a facilitated endocytosis mechanism [34]. Based on these results, it was concluded that the CSL nanoparticles exhibited greater activity than the free CSL. Furthermore, compared with PEG/CSL, PEG-C18/CSL, and TEG/CSL, the nanoparticles presented higher levels of antitumor activity (*p* < 0.05), owing to their different release rates. Based on the cumulative release curves, 70% of the CSL was released from PEG/CSL after 48 h of incubation, whereas approximately 50% of the CSL was released from PEG-C18/CSL or TEG/CSL nanoparticles. The free CSL molecules could not be effectively transferred through endocytosis, thus resulting in a low cytotoxicity of PEG/CSL.

### 2.6. Hemolytic Analysis

Based on our previous study, it was found that the CSL has severe membrane toxicity in red blood cells. To verify their levels of membrane toxicity, these CSL nanoparticles were incubated with red blood cell suspension for 4 h and detected with an Elisa reader, and their hemolytic rates are shown in Figure 7. Compared with the free CSL, the hemolytic rates of the CSL NPs were improved, which can be explained by the “core–shell” structure of the nanoparticles. After entrapping the CSL molecules into the core, the nanocarriers form the hydrophilic shell, through which they avoid or decrease the interactions between the CSL molecules and the erythrocyte membrane. Moreover, the hemolytic rates of all three CSL nanoparticles depended on the CSL concentration; no hemolysis was shown when the concentration was below 0.125 mg/mL CLS, while with a further increase in the concentration, these three CSL nanoparticles presented different hemolytic rates. PEG/CSL and PEG-C18 CSL revealed significant levels of membrane toxicity when the concentrations were 0.25 and 0.50 mg/mL, respectively, and the hemolytic rate exceeded 60%. For TEG/CSL, although membrane toxicity was observed when the concentration was 1.00 mg/mL, the hemolytic rate was below 20%. These results were consistent with those found for the cumulative release. A low release rate presented less opportunity for the free CSL molecules to interact with the red blood cell membrane, thus leading to a low hemolytic rate.

## 3. Materials and Methods

### 3.1. Materials

Celastrol (CSL, 98% purity) was purchased from Aktin Chemicals, Inc. (Chengdu, China). Tri(ethylene glycol) (TEG, Mn = 2217) and PEG-C18 were synthesized according to previous reports [35]. Polyethylene glycol (PEG, Mn = 2000) was obtained from Ponsure Biotechnology, Ltd. (Shanghai, China). The dialysis membrane (MWCO 8000~14,000) was purchased from Spectrum Laboratories Inc. (Los Angeles, CA, USA). The 4T1 cell line, cell culture materials, including 96-well plates, an RPMI 1640 medium, phosphate-buffered saline (PBS), 0.25% trypsin, and fetal bovine serum were obtained according to previous papers [36]. Acetonitrile and methanol were chromatographically pure and purchased from Scientific Fisher (Hampton, NH, USA). Other reagents were analytically pure and directly used without further purification.

### 3.2. CSL-Loaded Nanoparticles

The CSL (16 mg) and nanocarriers (PEG, PEG-C18, and TEG, 2 mg), were dissolved in 5 mL DMF, separately. According to a published paper [37], the organic solutions were injected into 5 mL of deionized water under continuous ultrasonication for 10 min, the mixture was added to the dialysis membrane, and deionized water (4 × 1 L) was used as the external dialysate, which was replaced every 2 h. These CSL-loaded nanoparticles with orange milky appearance were collected. The CSL concentration was detected via HPLC (Ultimate 3000, DIONEX, Sunnyvale, CA, USA) with a C18 column (5 µm, 4.60 mm × 250 mm); the specific conditions of detection were defined according to previous reports [38]. The calibration curve (Y = 0.5976 X − 1.5423, R^2^ = 0.9998) was utilized to calculate the CSL concentration. The DLC of the CSL in nanoparticles was calculated as follows:DLC% = (C_CSL_V/weight of nanoparticles) × 100%(1)

### 3.3. Dynamic Light Scattering (DLS)

The particle size, size distribution (polydispersity index, PDI), and zeta potentials were detected using a Zetasizer Nano-ZS analyzer (Malvern, UK) at room temperature. A backscattering detection pattern (He-Ne laser, 633 nm, 4 mV, and 173° scattering angle) was utilized to carry out the measurements. The samples (1 mg/mL) were measured three times.

### 3.4. Transmission Electron Microscopy

Transmission electron microscopy (TEM) measurements were performed with a JEM-1400 instrument at an accelerating voltage of 120 KV. The samples were prepared by drop-casting CSL-loaded nanoparticle solutions (0.1 mg mL^−1^) onto carbon-coated copper grids and dyed (2% uranyl acetate, *w*/*v*). After air-drying, the imaging of these samples was performed with TEM.

### 3.5. Stability Measurement

The samples were stored at 4 and 25 °C for 28 days. The particle sizes and PDI values were recorded using DLS at 0, 1, 2, 4, 7, 14, 21, and 28 days. The measurements were carried out in triplicate.

The media stability rates of these CSL nanoparticles were evaluated in a normal saline solution, a 5% glucose solution, a PBS solution (pH 7.4), and plasma. The CSL nanoparticle solution was mixed with 1.9% NaCl, 10% glucose, 2 × PBS, and plasma (1/4, *v*/*v*) to prepare the tested samples. The media stability rates were monitored for 24 h at 37 °C, and the particle size and PDI values of these samples were recorded at 0, 2, 4, 6, 8, 10, 12, and 24 h. The measurements were carried out in triplicate.

### 3.6. In Vitro Release

The CSL release profiles from the three nanoparticles were evaluated using the dialysis method. Briefly, 2 mL nanoparticle solutions were placed into a dialysis membrane and then immersed in 50 mL PBS. At a predetermined time, 5 mL PBS was withdrawn from the external solution, and fresh 5 mL PBS was added. The samples were measured with HPLC to determine the CSL concentration; the relative measurement conditions were explained in Section 3.2. Then, the cumulative release rate was calculated. The measurements were carried out in triplicate.

### 3.7. In Vitro MTT Assay

Briefly, 4T1 cells were cultured according to previous reports [39]. The cytotoxicity of the CSL nanoparticles was studied using an MTT assay. The 4T1 cells were seeded in a 96-well plate (8.0 × 10^3^ cells/well) at 37 °C with 5% CO_2_ for 24 h. The CSL nanoparticles were diluted to 0.25, 0.5, 0.75, 1, 1.5, 2, and 2.5 μg/mL with fresh media and then were added to each well to incubate with 4T1 cells. After 48 h incubation, an MTT solution (5 mg/mL, 20 μL) was added to each well and cultured for another 4 h. Subsequently, 150 µL dimethyl sulfoxide (DMSO) was added to the well after removing the medium. The optical density (OD) value of each well was recorded using an ELISA plate reader (Biotek, Winooski, VT, USA) at 570 nm wavelength. The half maximal inhibitory concentration (IC_50_) value was determined using the GraphPad Prism 5 software (No. 5.01). The cell inhibition rate was calculated as follows:Cell inhibition rate (%) = (1 − OD treated/OD control) × 100%(2)
where OD treated refers to the cells treated by the nanoparticles and CSL DMSO solution, and OD control refers to the cells treated using the RPMI-1640 medium.

### 3.8. Hemolytic Analysis

Briefly, 5 mL of the blood was drawn from the eyeball of Wistar rats. A red blood cell (RBC, 2% *w*/*v*) solution was prepared according to previous reports. The CSL nanoparticle solutions were incubated with 2% (*w*/*v*) RBC suspension at 37 °C for 4 h with different concentrations (0.06, 0.12, 0.25, 0.50, and 1.00 mg/mL, and CSL equivalent concentration). After centrifugation, 100 μL of the supernatant was placed into a 96-microwell plate, and the absorbance was measured at 540 nm using a microplate reader (Versamax Tunable Microplate Reader, Molecular Devices, San Jose, CA, USA). Deionized water and 0.9% NaCl were selected as the control, which was assumed to cause 100% and 0% hemolysis, respectively. The experiments were conducted in quintuplicate, and the data are shown as the mean values plus standard deviation (±SD). All experimental procedures were performed in accordance with the Ethical and Regulatory Guidelines for Animal Experiments as defined by the Animal Ethics Committee of Peking Union Medical College (Beijing, China).

### 3.9. Statistical Analysis

The data between groups were compared using one-way analysis of variance (ANOVA) (SPSS 25.0, Chicago, IL, USA), and *p* < 0.05 indicated statistical significance.

## 4. Conclusions

To evaluate the structural effects of nanocarriers, PEG, PEG-C18, and TEG were utilized to entrap the hydrophobic anticancer drug CSL. An injection-combined antisolvent precipitation method was used to effectively prepare the three CSL nanoparticles, and orange milky solutions were obtained. These CSL nanoparticles presented small particle sizes of approximately 120 nm and narrow polydispersity indexes. After storage for 28 days, the CSL nanoparticles showed excellent stability, and their particle size, size distribution, and zeta potential were maintained. In addition, all of these CSL nanoparticles presented good media stability, and no significant change was shown in PBS, 5% glucose solution, and plasma. Based on their different release profiles, the CSL nanoparticles exhibited different levels of antitumor activity in vitro and membrane toxicity in red blood cells. Compared with linear PEG/CSL nanoparticles, PEG-C18/CSL and TEG/CSL nanoparticles revealed slow release rates, high antitumor activity, and low membrane toxicity. Especially for TEG/CSL nanoparticles, the CSL release process could sustain for 8 days, and the hemolytic rate was below 20%. In conclusion, the structural effects of nanocarriers can affect the interactions between nanocarriers and drugs, resulting in different release profiles and antitumor activity levels. These influencing factors should be systematically considered to prepare an ideal drug delivery system.

## Figures and Tables

**Figure 1 molecules-28-01040-f001:**
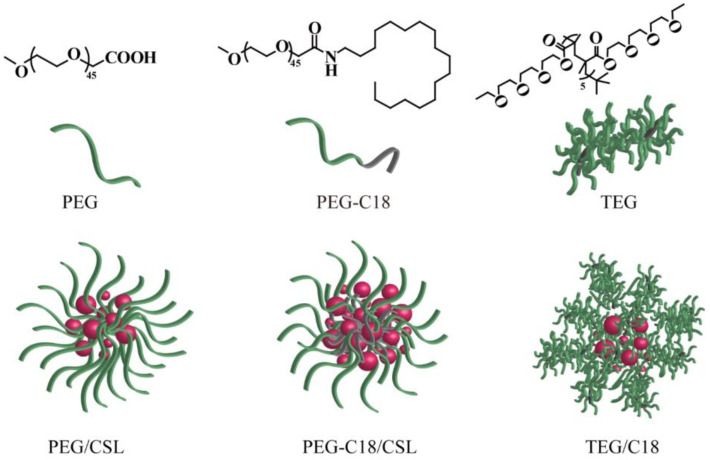
Structure of three nanocarriers and illustration of CSL-loaded nanoparticles.

**Figure 2 molecules-28-01040-f002:**
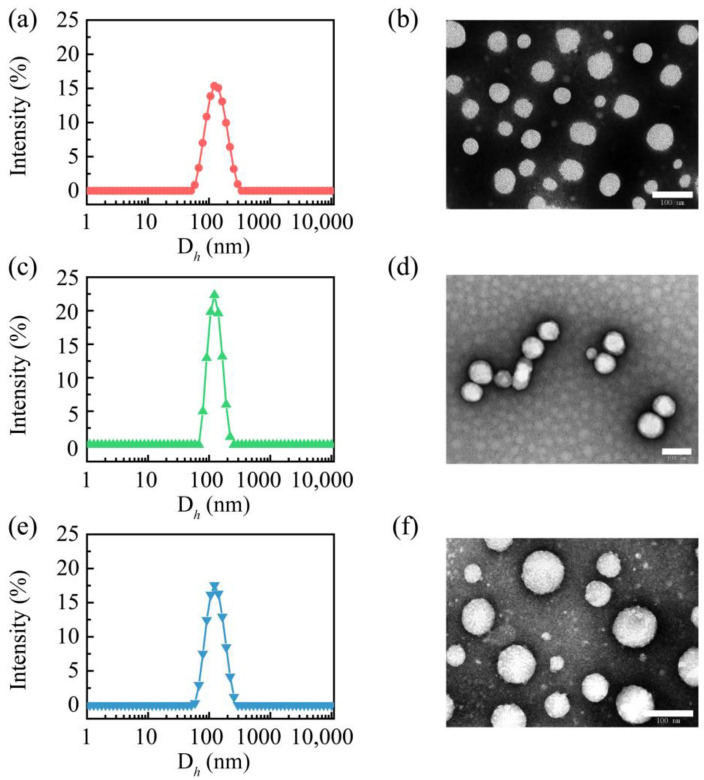
Particle size distribution curves of PEG/CSL (**a**), PEG-C18/CSL (**c**), TEG/CSL (**e**), and TEM images of PEG/CSL (**b**), PEG-C18/CSL (**d**), and TEG/CSL (**f**). Scale bar: 100 nm.

**Figure 3 molecules-28-01040-f003:**
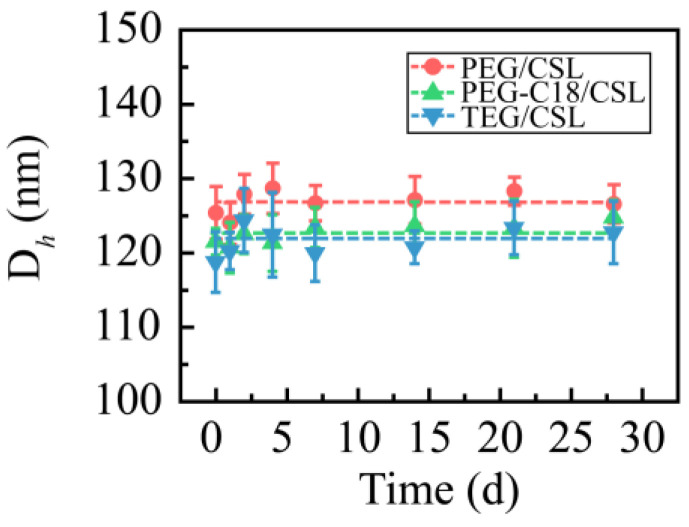
Particle size of three CSL nanoparticles during the entire storage process (n = 3).

**Figure 4 molecules-28-01040-f004:**
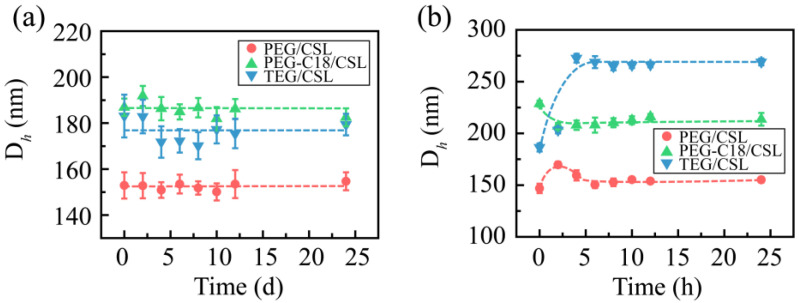
Particle sizes of CSL nanoparticles in 5% glucose solution (**a**) and plasma (**b**) after 24 h of incubation, n = 3.

**Figure 5 molecules-28-01040-f005:**
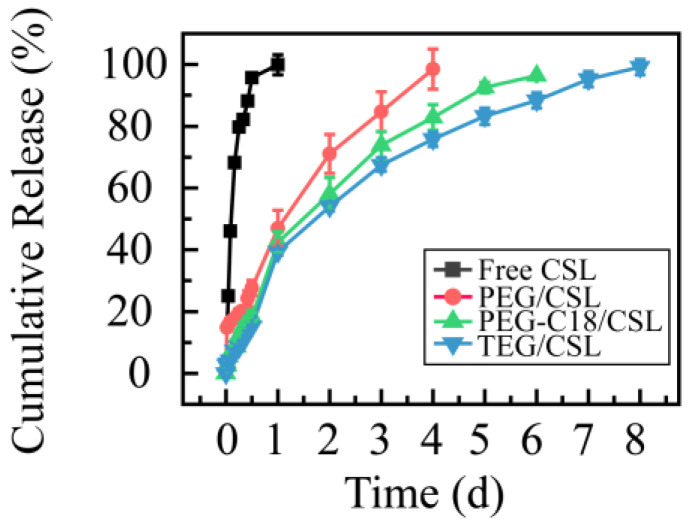
Cumulative release rates of CSL DMSO solution and CSL nanoparticles at 37 °C, n = 3.

**Figure 6 molecules-28-01040-f006:**
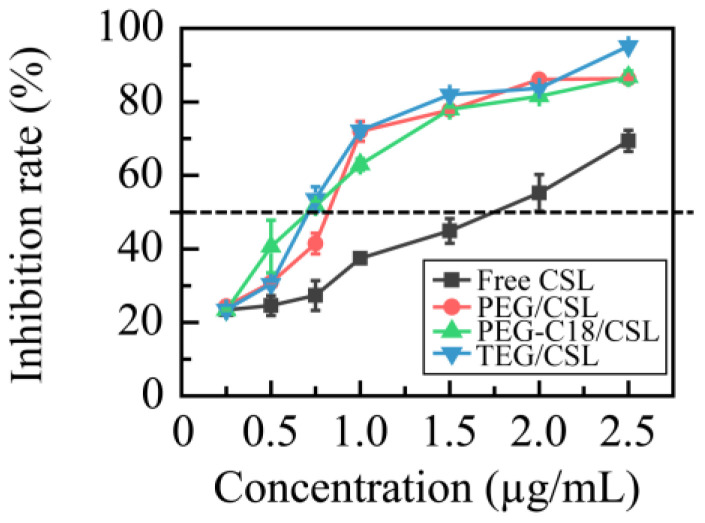
Inhibition rates of free CSL and CSL nanoparticles in 4T1 cell line at 37 °C after 48 h incubation, n = 5.

**Figure 7 molecules-28-01040-f007:**
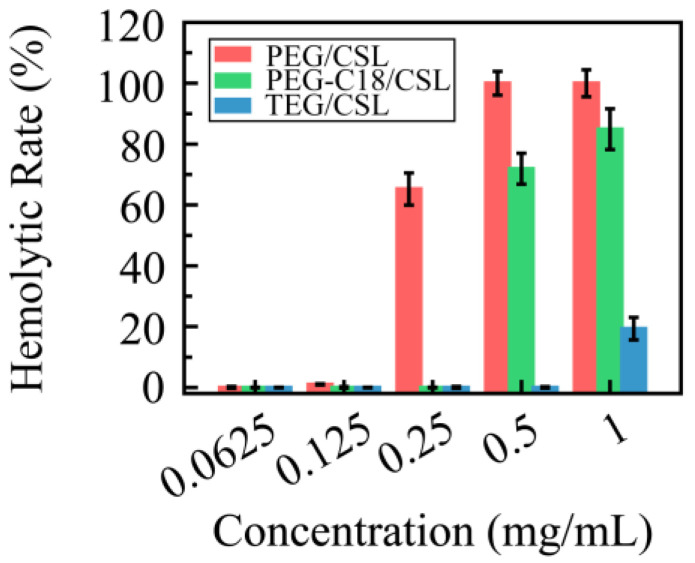
Hemolytic rate of CSL nanoparticles in normal red blood cells at 37 °C after 4 h incubation, n = 5.

**Table 1 molecules-28-01040-t001:** Results of three CSL-loaded nanoparticles.

Samples	DLS Results			HPLC Results	
	D*_h_* (nm) ^a^	PDI	ζ (mV) ^b^	EE (%)	DLC (%)
PEG/CSL	125.4 ± 3.2	0.10 ± 0.01	−23.7 ± 0.6	44.3 ± 5.6	77.8 ± 3.2
PEG-C18/CSL	121.7 ± 1.3	0.09 ± 0.04	−23.2 ± 0.4	83.7 ± 3.5	87.3 ± 2.8
TEG/CSL	119.0 ± 0.9	0.11 ± 0.01	−25.7 ± 0.6	26.9 ± 4.9	68.5 ± 3.1

^a^ Hydrodynamic diameter. ^b^ Zeta potential. All the data are mean value ± SD, n = 3.

## Data Availability

Not applicable.

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
