# Peer review of "A Celastrol Drug Delivery System Based on PEG Derivatives: The Structural Effects of Nanocarriers"

_molecules, 2023, doi:10.3390/molecules28031040_

Round 1

Reviewer 1 Report

The authors conclude "To evaluate the effects of hydrophobicity and steric hindrance, three nanocarriers in-cluding PEG, PEG-C18, and TEG were utilized to entrap hydrophobic anticancer drug CSL."  - but no mechanism or experiemnts in this regard have been provided nor any bio work images.  

hydrophobic effects and steric hindrance can affect the interactions - proof ?

The samples were measured by HPLC to determine the con-centration of CSL; and then, the cumulative release rate was calculated. The measure-ments were carried out in triplicate - no HPLC profile data is presented ?

 Based on our previous study, it was found that CSL showed severe membrane toxicity towards blood red cell - what is the mechanism of change ?

  1. Page no: 1 ,16 th line spelling mistake marked in yellow color
  2. page no:5 First paragraph 2 nd line Specify the solution 
  3. page no:6 1st Para 11th line check the sentence
  4. page no:6 13th line mention hours highlighted in yellow color 
  5. page no: 8 2nd Para 1st  line grammatical mistake- (storage-stored)
  6. page no:9 in conclusion grammatical mistake  (storage-stored)
  7. page no: 9 15th line spelling mistake highlighted in yellow color

Reviewer 3 Report

Manuscript Review Molecules 2095663

 Title: Drug delivery system based on derivatives of PEG: Structural effects of nanocarriers

 This manuscript focuses on the development of celastrol-loaded nanoparticles using three analogues including polyethylene glycol (PEG), PEG conjugating octadecylamine (PEG-C18), and tri(ethylene glycol) (TEG) via injection combining dialysis method. This formulation is homogenous and stable. Such systems attract attention by their great potential for cosmetic, food, and nanomedicine applications. The work needs minor revision with an additional calculation.

 Minor comments

 1. Article title is rather broad. It is better to be specific, because this paper is predominantly about formulations only for celastrol (CSL)-loaded nanoparticles via injection combining dialysis method.

2. Page 2. “To compare and study the structural and component effects of nanocarriers system-atically, in this study, three different PEG derivatives are synthesized and applied as nanocarriers to prepare NDDS, in which hydrophobic celastrol is selected as model drug”. It is better to keep only “applied as nanocarriers” because Tri(ethylene glycol) (TEG, Mn = 2217) and PEG-C18 was synthesized according to previous reports [35].

3. Page 2, section 2.1 “DLC of these CSL nanoparticles were detected via HPLC. All of these nanocarriers presented moderate to good encapsulated efficacy”, it is better to use drug loading capacity (DLC). Could you, please, calculate the encapsulate efficiency (EE, %).

4. Page 3. Section 2.2 (Table 1). Particle characteristics (size, zeta potential) and morphology (TEM results) for three different PEG derivatives are very similar. This is interesting, since used PEG derivatives are with different hydrophilic-lipophilic balance. Can you explain these results? Did the authors try to obtain empty nanoparticles via injection combining dialysis method? Is there a difference between empty and celastrol-loaded nanoparticles?

 5. I wouldn't claim about “the steric hindrance hampers” or “steric hindrance of TEG/CSL” in Conclusion and Abstract, because other specific methods to confirm are need it.

 6. Page 7. Section 3.2.” CSL (16 mg) and nanocarriers (2 mg), including PEG, PEG-C18, and TEG, were dissolved in 5 mL DMF to prepare the organic phases”. What are these nanocarries? There were prepared in advance? Or they are used like as substances?

 7. Page 7. Section 3.2 the references about the injection combining dialysis method of nanoparticle preparation are need it. Is it possible to use other organic solvents or only DMF?

Round 2

Reviewer 1 Report

The authors have satisfactorily revised though bio related information is missing.